# The Role of Inflammation in the Pathogenesis of Cardiogenic Shock Secondary to Acute Myocardial Infarction: A Narrative Review

**DOI:** 10.3390/biomedicines12092073

**Published:** 2024-09-11

**Authors:** Irina Kologrivova, Maria Kercheva, Oleg Panteleev, Vyacheslav Ryabov

**Affiliations:** 1Cardiology Research Institute, Tomsk National Research Medical Center, Russian Academy of Sciences, 111A Kievskaya, Tomsk 634012, Russia; panteleevoo@cardio-tomsk.ru (O.P.); rvvt@cardio-tomsk.ru (V.R.); 2Cardiology Division, Siberian State Medical University, 2 Moscovsky Trakt, Tomsk 634055, Russia

**Keywords:** cardiogenic shock, myocardial infarction, myocardial infarction-associated cardiogenic shock, pathophysiology, inflammation, monocytes, neutrophils, clonal hematopoiesis, anti-inflammatory therapy, hemoadsorption

## Abstract

Cardiogenic shock (CS) is one of the most serious complications of myocardial infarction (MI) with a high mortality rate. The timely and effective prevention and early suppression of this adverse event may influence the prognosis and outcome in patients with MI complicated by CS (MI CS). Despite the use of existing pharmaco-invasive options for maintaining an optimal pumping function of the heart in patients with MI CS, its mortality remains high, prompting the search for new approaches to pathogenetic therapy. This review considers the role of the systemic inflammatory response in the pathogenesis of MI CS. The primary processes involved in its initiation are described, including the progression from the onset of MI to the generalization of the inflammatory response and the development of multiple organ dysfunction. The approaches to anti-inflammatory therapy in patients with CS are discussed, and further promising research directions are outlined. In this review, we updated and summarized information on the inflammatory component of MI CS pathogenesis with a particular focus on its foundational aspects. This will facilitate the identification of specific inflammatory phenotypes and endotypes in MI CS and the development of targeted therapeutic strategies for this MI complication.

## 1. Introduction

Cardiogenic shock (CS) represents a critical emergency situation that may arise as a complication of underlying medical conditions, such as acute coronary syndrome, acute myocarditis, decompensation of chronic heart failure (HF), progression of acute HF, intractable ventricular arrhythmias, and severe valve pathology [1]. This condition is characterized by inadequate cardiac output, failing to keep up with the metabolic demands of the body, which in turn results in the life-threatening hypoperfusion of internal organs (heart, brain, kidneys, lungs, liver, and intestines) [2]. The in-hospital mortality rate among patients with CS is 30–60% with the majority of deaths occurring within the first 24 h of admission. The long-term prognosis for patients with CS is similarly poor with a 1-year mortality rate of up to 50–60% [3].

The incidence of myocardial infarction complicated by CS (MI CS) is estimated to be 30–40% of the total number of patients with CS [4]. Despite the timely and complete myocardial revascularization, modern devices of mechanical circulatory support (MCS) and existing advances in pharmacotherapy, the incidence of this complication, as well as its fatal outcome, has remained consistently high over the past decade. It is likely that achieving and maintaining optimal myocardial pumping function is not the only optimal way to stabilize the condition and improve the prognosis in this cohort of patients. High incidence rates suggest the necessity for further investigation into potential novel therapeutic targets associated with the development and progression of MI CS [5,6,7]. For a long time, the staging system for CS severity lacked uniform criteria, which impeded a more comprehensive understanding of the progression of this life-threatening complication of cardiovascular pathology. The staging system for CS severity developed by the American Society for Cardiovascular Angiography and Interventions (SCAI) in 2019 differentiated stages of cardiogenic shock from A to E. Stage A is “at risk” for CS, stage B is “beginning” shock, stage C is the “classic” CS, stage D is “deteriorating”, and E is “extremis”, which is highly unstable, often with cardiovascular collapse [8]. Despite the recognition of shock stages and the definition of several clinical characteristics, there are likely additional, previously unidentified mechanisms of MI CS pathogenesis. The consideration of underlying pathogenic mechanisms will facilitate the development of targeted therapeutic approaches for this patient cohort, potentially altering the course and prognosis of this condition.

One of such pathogenic mechanisms of development and progression of MI CS is activation of the systemic inflammatory response [3]. Due to the heterogeneity of patients with MI CS, systemic inflammatory response syndrome (SIRS) frequently occurs through the aseptic inflammatory response that arises due to ischemia and the septic inflammatory response that arises from the accompanying comorbid (infectious) background [9]. Nowadays, there is no understanding of how septic and aseptic inflammatory components contribute to the development and progression of shock; specific markers have not been identified to distinguish aseptic inflammation from septic inflammation in individuals with comorbid pathology [5,6]. The inflammatory response is typically more pronounced in MI CS patients, although it does not reach the level observed during septic shock [5]. Some data indicate that in approximately 6% of cases, patients with CS may develop sepsis, which significantly contributes to the severity of inflammation, changes it from aseptic to septic shock and markedly worsens the prognosis in this cohort of patients [10]. Furthermore, multiple organ dysfunction and impaired intestinal and pulmonary barrier function result in a mixed inflammatory phenotype in patients with CS, which differs from that observed in MI patients without CS [6]. The pivotal role of inflammation in the pathogenesis of CS is further supported by encouraging, but not yet comprehensive, outcomes of hemoadsorption and anti-inflammatory therapy in patients with and at high risk of CS [11].

Apparently, the identification of the predominant stage at which dysfunction of the inflammatory response develops in a patient with MI CS may facilitate the implementation of anti-inflammatory therapy in MI CS patients in a more targeted and personalized manner, enhancing its efficacy. The development of systemic inflammation in patients with MI CS commences at the myocardial injury stage. Therefore, it is crucial to assess the role of all major cellular and molecular effectors at this initial stage.

The aim of the present narrative review was to update and summarize the existing data on the inflammatory component of MI and MI CS pathogenesis, its fundamental and translational aspects, and to provide information on the completed or ongoing clinical trials aiming to control inflammation during MI CS.

## 2. Methods

The following databases were used to extract the articles relevant to the topic: PubMed; Google Scholar; Scopus; Russian Index of Science Citation. The following keywords and their combinations were used during the literature search: “cardiogenic shock”; “myocardial infarction”; “inflammation”; “cardiac intensive care unit”; “immune modulatory therapy”; “cytokine sorption”; “hemoadsorption”; “endophenotype”; “inflammatory phenotype”; “inflammatory endotype”; “neutrophils”; “NETosis”; “monocytes”; “macrophages”; “lymphocytes”; “T-lymphocytes”; and “clonal hematopoiesis”. Both original articles and critical reviews were selected for analyses. Critical reviews were selected only if access to the full text was available. Original articles were selected if access to either full text or abstract was available. No language restriction was applied. Works of the recent 5 years were primarily analyzed. If any seminal paper older than the recent 5 years was mentioned in the critical review, the reference to the original article was provided.

## 3. Inflammatory Response in Acute Myocardial Infarction

MI serves as a substrate for the development of CS in the majority of cases [12]. It occurs when the myocardium does not receive sufficient oxygen supply to correspond to its demands, which may happen due to various reasons, among which the most prevalent are the rupture or erosion of atherosclerotic plaque, coronary embolism, and coronary microvascular dysfunction [13,14]. As a result, injury of the heart tissue takes place, which is followed by the loss of cardiomyocytes and other cells of the myocardium through different types of cell death: necrosis, apoptosis, necroptosis, pyroptosis, ferroptosis, or autophagy [15,16,17]. Further on, both the systemic and the local inflammatory responses are initiated [15].

Inflammation plays an integral role in the pathogenesis of MI [18]. The inflammatory response is required to induce a regenerative response following myocardial injury and initiate post-infarction scar formation [19]. Traditionally, the inflammatory response after MI is divided into three stages: the early inflammatory stage (72 h after MI), the late reparative and proliferative stage (7–10 days after MI), and the maturation stage (lasting for six months or even longer) [20]. Patients with MI CS are at risk of developing an excessive inflammatory response, which can lead to the generalization of inflammation and dysregulation of repair mechanisms [21]. The responses evoked by inflammation are a keystone of pathology, posing a threat to the patient’s own tissues and organs.

In response to cellular damage, danger-associated molecular patterns (DAMPs)—molecular danger signals that can be recognized by specialized receptors (pattern recognition receptors, PRR) on the surface of monocytes, macrophages, and neutrophils—are released into the bloodstream. Classic DAMPs in acute myocardial infarction (MI) include high-mobility group protein B1 (HMGB1, amphoterin), heat shock proteins, s100 proteins, and nucleic acids, including mitochondrial DNA [22,23,24] (Figure 1). DAMP–PRR binding initiates an intracellular signaling cascade and activates the nuclear factor (NF)-κB and mitogen-activated protein kinase-dependent signaling pathways, which are the primary initiators of synthesis of pro-inflammatory cytokines, chemokines, and adhesion molecule expression [25]. In addition to immune cells, DAMPs are also recognized by receptors expressed on fibroblasts, cardiomyocytes and endothelial cells, which results in their apoptosis and increased cardiac tissue injury [26]. Furthermore, the complement system may be activated through DAMPs, which could contribute to inflammation development (Figure 1). The complement components C3a and C5a are potent chemoattractants and facilitate the recruitment of neutrophils to the infarction zone [27].

When the danger signal is received, cells of the immune system are recruited to the myocardium to remove degradation products from cardiac tissue (Figure 1). First, monocytes (in the first 30 min) and then neutrophils (in the first hours) migrate to the infarct zone under the influence of DAMPs and chemokines [31]. The migrated monocytes differentiate into macrophages and replace the resident myocardial cells [32]. However, the activation of granulocytes, monocytes, and macrophages associated with the production and secretion of pro-inflammatory cytokines and bioactive substances, and the release of neutrophil extracellular traps (NETs), causes additional damage to the myocardium [33]. Neutrophils release NETs, become sources of DAMPs, and activate NLRP3 inflammasomes in macrophages. At the same time, they promote macrophage polarization toward an anti-inflammatory M2 phenotype [34]. Thus, the balance between pro- and anti-inflammatory mechanisms of innate immunity comes to the fore in MI-associated inflammation.

In addition to the local inflammatory response, MI triggers hematopoiesis in the bone marrow and extramedullary hematopoiesis in the spleen [21,35,36].

Myocardial injury is also associated with the recruitment of adaptive immune cells, T and B lymphocytes, which peak on days 5–7 after MI [28] (Figure 1). These are typically de novo recruited peripheral blood cells. Among T lymphocytes, both inflammatory CD8+ cytotoxic cells, CD4+ T helper type 1 (Th1), and regulatory FoxP3+ T lymphocytes are detected, whereas Th2 and Th17 subpopulations are almost undetectable in postinfarction myocardium [28,37]. The imbalance between Th17 and T regulatory lymphocytes can be observed in the periphery [38]. Cytotoxic CD8+ T lymphocytes can promote the development of inflammation by inducing cardiomyocyte apoptosis, switching macrophages to an inflammatory phenotype, and secreting damaging agents during degranulation [29]. On the other hand, their activation is essential for the regulation of the activity of myocardial fibrosis and remodeling [39]. Th1 lymphocytes have a similar function, contributing to the balance between inflammation and cardiac tissue repair, mainly through the production of cytokines interferon γ (IFN-γ), IL-6 and TNF (Yuan D, 2019) [40]. A unique subpopulation of regulatory T lymphocytes (Treg) possesses exceptional anti-inflammatory activity: Tregs minimize cardiac tissue injury and prevent cardiomyocyte apoptosis and excessive fibrosis development [41]. The high prognostic significance of the neutrophil–lymphocyte ratio in patients with ST-segment elevation MI may be primarily due to the important regulatory potential of lymphocyte subpopulations [42].

The contribution of microRNAs to the regulation of MI-associated inflammation has been also shown. Thus, miR-146a-5p, miR-155, miR-22 potentiate the inflammatory response, whereas miR-21, miR-133a, miR-19a/19b suppress the activity of inflammatory signaling pathways [30].

Atherosclerosis is known to be accompanied by the age-related low-grade chronic inflammation, which is characterized by immune senescence, cellular senescence and autophagy defects. A special term “inflammaging” was coined to determine this condition [43]. Inflammaging has been demonstrated to influence cardiovascular health and the course and outcomes of MI in particular [44]. The damage of cardiomyocytes leads to the activation of the kinases, activating the process of cellular senescence and activating a senescence-associated phenotype (SASP) [15]. Mice that were given a senolytic drug, navitoclax, before the ligation of the left anterior descending artery presented with less pronounced inflammation, better survival and improved cardiac function [45]. The inhibition of senescence in cardiomyocytes limited the degree of inflammaging, ameliorated cardiac function and limited heart injury in mice [46]. Inflammaging may also impact extracellular matrix degradation [47], potentially leading to the more pronounced impairment of cardiac output.

Ultimately, the imbalance between inflammation and its resolution, which may be based on genetic, epigenetic, infectious and metabolic mechanisms, tends to generalize the inflammatory process, which can cause significant damage to the myocardium and impairment of myocardial function. A sharp decrease in left ventricular contractility results in a critical decrease in mean arterial pressure and cardiac output, and systemic hypoperfusion develops, leading to multiple organ dysfunctions. In addition, the compensatory activation of baroreceptors and chemoreceptors in response to decreased arterial pressure leads to increased arteriolar tone, which further increases visceral hypoperfusion. This results in the SIRS, systemic vasodilation, a further decrease in blood pressure, and an exacerbation of the inflammatory response in the myocardium, which becomes maladaptive [9]. MI CS develops typically 6 h after the onset of MI [48]. This creates a vicious cycle whereby multiple organ dysfunction and inflammation potentiate each other. Other mechanisms of MI CS development in MI patients include mechanical complications such as ventricular septal defects, acute mitral regurgitation due to papillary muscle rupture, pseudo-aneurysm development, and left ventricular free-wall rupture [49].

With mechanical and pharmacological circulatory support, the inflammatory component of MI CS development appears to be unrecognized and may contribute to compromised hemodynamics. Being unrecognized in short-term CS survivors, SIRS may have adverse effects when these patients are followed for 3 or 6 months or longer.

## 4. Clinical Studies Investigating Inflammation in Cardiogenic Shock

Studies into the inflammatory response in patients with CS are mostly retrospective or involve a small number of patients. Table 1 summarizes a far from complete list of studies investigating the impact of inflammation on prognosis in patients with CS. In general, all studies indicate that SIRS is associated with a worse prognosis in patients, and it does not always correlate with the development of secondary infection. In addition, all authors face similar limitations: the heterogeneity of patients in terms of the time of the first blood collection for analysis and the inability to study inflammatory markers in patients who did not survive to day 7 or even day 3 of follow-up, which can cause distorted results and lead to the wrong conclusions. In addition, patients with CS typically have comorbidities aggravating short- and long-term mortality that further complicate studies aimed at investigating the pathogenesis of CS. The most common comorbidities include arterial hypertension, diabetes mellitus, and chronic kidney disease [50].

In a prospective study, an attempt to evaluate cellular markers of inflammation, including major leukocyte subpopulations, was made by a group of scientists from Switzerland led by Lucas Liaudet (2020) [51]. An increased innate immune response at day 1 of CS development, manifested by neutrophilia and an increased production of IL-6, IL-10, and Monocyte Chemoattractant Protein-1 (MCP-1), correlated with the severity of CS and was followed by the development of immune insufficiency in the form of decreased levels of monocytes and lymphocytes at days 3 and 7. High eotaxin levels at day 1 were accompanied by the development of eosinophilia, which can be considered a compensatory mechanism initiating tissue regeneration. Most patients developed secondary infections; however, the immune system did not respond adequately in the presence of severe multiple organ dysfunctions, which also supports the hypothesis of the development of the so-called “immunoparalysis” in severe CS [51].

High levels of acute phase proteins (CRP, pentraxin-3 (PTX-3), procalcitonin, presepsin) in patients with MI-CS at day 1 after admission to hospital were associated with increased 3-month mortality regardless of the presence or absence of secondary infection. Nevertheless, the associated infectious process affected the dynamics of inflammatory biomarkers: the main difference was a prolonged increase in the level of presepsin, which reached its peak after 48 h of follow-up, with no significant dynamics of PTX-3 in patients with respiratory infection [5].

A large retrospective study showed that the cumulative index of the inflammatory response, calculated based on the level of high-sensitivity CRP (hsCRP) and total leukocyte count on days 2–3 of follow-up in patients with MI CS, was inversely correlated with left ventricular ejection fraction and directly correlated with the lactate level and SCAI severity. Patients with the most severe inflammation showed significantly increased 30-day mortality [52]. 

Recent data from the French FRENSHOCK study show that even an isolated assessment of CRP levels on admission to hospital, independent of other inflammatory markers, can identify a cohort of patients with CS at high risk of death after both 1 month and 1 year of follow-up [53].

According to the data from the registry of the University Center for Cardiovascular Diseases in Hamburg, each 50 mg/L increase in the CRP level was associated with an 8% elevated risk of 30-day mortality in patients with CS. However, this correlation was not observed when MCS (venoarterial extracorporeal membrane oxygenation (ECMO)) or the «Impella» support system was used [54]. However, the CRP level did not depend on the severity of CS according to the SCAI scale.

In another study, which enrolled only patients with CS and MCS, a lower neutrophil to lymphocyte ratio was a predictor of short-term survival and correlated with the severity of CS according to the SCAI scale, and the successful use of MCS led to decreased IL-6 levels [55].

The largest retrospective study (approximately 9000 patients enrolled) of SIRS in patients with CS was conducted at the Mayo Clinic [3]. It was shown that patients with SIRS are characterized by a more severe course of CS, a more severe multiple organ dysfunction, and an increased risk of both short- and long-term mortality.

A recent multicenter study showed a fundamental difference between the dynamics of biomarkers reflecting inflammation and neurohumoral status in patients with CS and in patients with septic shock. Adrenomodulin, a biomarker of generalized cardiovascular stress, neurohumoral activation and inflammation, that contributes to vasodilation, was found to be elevated in both groups of patients, but the correlation of its elevated levels with mortality was demonstrated only in CS patients [56]. Furthermore, patients with CS were characterized by a slower increase in the CRP level compared to patients with septic shock [56].

## 5. Pathogenesis of Inflammation Development in Myocardial Infarction-Associated Shock

From a physiological perspective, the inflammatory response triggered by cardiomyocyte necrosis and tissue hypoxia in MI CS performs a protective and reparative function; however, in some cases, this response becomes dysregulated and maladaptive, leading to injury [6]. Consequently, an excessive inflammatory response impairs microcirculation, causes excessive vasodilation and exacerbates cardiac cell death, which may subsequently provoke shock progression.

At the same time, SIRS severity varies among patients with acute MI and does not depend on infarct size. This suggests the presence of the other key regulatory and signaling pathways that can be therapeutically affected [57].

Aseptic inflammation is typically a secondary response to extensive myocardial damage caused by infarction [58]. In addition, there is evidence that ECMO and other invasive MCS methods can act as modulators of the activity and number of immune cells. In general, the decreased number and functional inhibition of lymphocytes and monocytes was accompanied by an increased number and enhanced activation of neutrophils [59]. As previously stated, a vast range of DAMPs can trigger inflammatory responses in CS [22]. During MI CS development, intestinal permeability markedly increases, facilitating the penetration of bacterial components into the systemic bloodstream, the emergence of endotoxinemia, and the onset of infection [60] (Figure 2). Thus, MI CS development provides an additional signal for innate immunity activation in the form of pathogen-associated molecular patterns (PAMPs). Inflammation intensifies and triggers the risk of depletion of its protective potential and accession of secondary infection.

A number of factors contribute to the development of a respiratory infection in patients with MI CS within the first 48 h after admission to hospital. These include pulmonary congestion, mechanical ventilation measures, pre-hospital cardiac arrest using cardiopulmonary resuscitation methods, therapeutic hypothermia, and invasive MCS methods [5]. The incidence of sepsis in patients with CS varies from 6 to 50%, depending on the scale of the study (in small-scale studies, the incidence was higher) [6]. In any case, septic inflammation exacerbates MI CS severity and increases the risk of multiple organ dysfunction and mortality. Nevertheless, it remains unclear whether infectious complications are the primary drivers of immune incompetence or develop secondary to immune disorders and aseptic inflammation [5]. Apparently, the identified primary inflammatory associations in MI CS pathogenesis may help to identify additional endophenotypes [5,6]. Based on the endophenotype identified, measures to prevent MI CS development in patients at risk for shock and management of patients with MI CS can be targeted.

One of the key drivers behind the development of CS is the intracellular enzyme dipeptidyl peptidase 3 (DPP3), which is a zinc-dependent serine protease that cleaves dipeptides from N-terminus of tetra- or decapeptides, exerting a regulatory function. When secreted outside the cell, DPP3 acts as an important regulator of the activity of the renin-angiotensin–aldosterone system. In CS, high levels of circulating DPP3 facilitate an excessive inactivation of angiotensin II, which contributes to the maintenance of hypotension resistant to angiotensin II therapy [61]. An increase in the DPP3 level is associated with the increased severity of CS, as well as a higher incidence of target organ dysfunction, including acute renal failure, refractory shock, and in-hospital mortality [51]. In a mouse model of acute HF, the administration of a monoclonal antibody to DPP3 (Procizumab) normalized cardiac function, improved renal hemodynamics, reduced the severity of oxidative stress, and suppressed an inflammatory response [62].

CRP, the most studied inflammatory marker, is elevated in response to increased IL-6 levels in CS as part of SIRS. Its concentration above 10 mg/L is pathological [63]. Similar to C1q, CRP, a short PTX, activates complement via the classical pathway. On the one hand, this has a protective effect, facilitating the removal of damaged cells. On the other hand, it promotes monocyte infiltration of the myocardium and other organs damaged by CS [5]. It was shown that CRP significantly increases the activity of NO synthase [64], which can potentially exacerbate the severity of hypotension in patients with CS.

Pentraxin 3 (PTX-3), a long PTX, is a more specific marker of inflammation associated with atherosclerotic damage. Its release kinetics is similar to that of IL-6, and its concentration increases in proportion to infarct size [65]. The prognostic significance of PTX-3 as a predictor of 3-month mortality in patients with CS was demonstrated [5]. However, PTX-3 is likely to act as a biomarker and does not have an additional pathological effect in CS. In contrast, PTX exhibits anti-inflammatory properties. It inactivates the C1q component of the complement system, reduces neutrophil infiltration into tissues, and controls platelet activation and thrombosis development [5,65].

In patients with MI CS, the increased level of procalcitonin, a precursor of the thyroid hormone calcitonin, may indicate a bacterial infection and is not associated with myocardial damage [66,67]. The current evidence is insufficient to support the assertion that procalcitonin can be considered an independent prognostic indicator of mortality in CS [68].

In addition, the concentration of the secretory form of the CD14 (sCD14), also known as presepsin, was shown to increase in CS. Presepsin was previously considered a marker of bacterial infection, but subsequent research demonstrated that its concentration increases in all conditions associated with increased MCP-1 production and monocyte activation [59]. Earlier studies demonstrated that presepsin exerts regulatory functions, suppressing T-lymphocytes activation and IFN-γ production [69]. On the other hand, it was shown to enhance the stimulation of endothelial cells by lipopolysaccharides, which facilitated the synthesis of inflammatory cytokines, triggered the release of matrix metalloproteinase 9 from granulocytes, and increased the expression of adhesion molecules on blood cells [70].

The above humoral inflammatory factors ultimately help activate or control the function of innate or adaptive immune cells. Different subpopulations of immune cells contribute to the generalization of inflammation and the development of organ dysfunction. A detailed description of all cell subpopulations in the context of CS pathogenesis remains incomplete. The aim of this study was to provide an overview of the available data on the contribution of different immune cells to inflammation in MI CS.

## 6. Macrophages in Myocardial Infarction-Associated Shock

In the early stages of acute MI, monocytes are recruited to the infarction zone via the chemokine CCL2 (C-C motif ligand 2), which is more commonly referred to as MCP-1 (Figure 3). The excess level of MCP-1 can induce apoptosis in intact cardiomyocytes [71] and is an unfavorable prognostic indicator in patients with MI CS [72]. Unexpectedly, the administration of hypothermia in patients with CS resulted in a significantly elevated level of MCP-1 during the heating phase, thus conferring a favorable prognostic outcome [71]. The protective effect associated with elevated MCP-1 levels may be due to the activation of macrophage subpopulations that possess reparative properties. Among patients with septic cardiomyopathy, the level of CD163+ Resistin-like alpha (RETNLA)+Mac1 macrophages in cardiac tissue, which exhibit a high expression of TREM2 (triggering receptor expressed on myeloid cells 2), was identified as a critical factor affecting the prognosis of patients with septic shock. This macrophage subset was responsible for the clearance of mitochondria released during myocardial injury, promoting the restoration of local homeostasis in septic shock [31]. Whether this macrophage subset is involved in MI CS progression has yet to be explored.

## 7. Neutrophils in Myocardial Infarction-Associated Shock

It has recently been demonstrated that NETs play an important role in the formation of arterial thrombosis [73]. NETs represent neutrophil DNA, which, in the form of a reticular network, enters the extracellular environment upon cell activation and the initiation of the NETosis mechanism. Consequently, the DNA is linked to a multitude of neutrophil enzymes and biologically active substances, including neutrophil elastase, myeloperoxidase, and neutrophil cationic proteins [74]. During an infectious process, NETosis performs a protective function and contributes to a more effective elimination of the infectious agents. However, in patients with cardiovascular pathologies, the increased ability of neutrophils to release DNA becomes pathological as aseptic inflammation develops. This can not only be an unfavorable biological marker but also a factor that exacerbates the course of the disease (Figure 3). Among patients with acute MI, the release of NETs was observed to be higher in the infarct-related artery compared to peripheral arteries. In addition, the intensity of NETosis was found to correlate with the incidence of in-hospital adverse cardiovascular events [75]. Another study demonstrated that patients with MI CS exhibited an increase in NETosis markers, circulating double-stranded DNA, and citrullinated histone H3. However, the level of these markers did not correlate with infarct size. Apparently, the triggering of NETosis in CS is associated with systemic hypoperfusion, hypoxia, and target organ damage [75].

On the other hand, the elevated level of the cellular stress-associated cytokine growth-differentiation factor 15 (GDF-15) was identified as a predictor of 30-day mortality in patients with CS. Furthermore, this factor was included into a multiple regression model for risk prediction in patients with CS along with other parameters such as age, lactate level, left ventricular ejection fraction, and TIMI (thrombolysis in myocardial infarction) blood flow velocity < 3 [76]. GDF15 is known to inhibit the recruitment of neutrophils to the site of inflammation, which can impair their physiological function, namely the resolution of inflammation and infection [77]. The adequate activation of neutrophils is likely to be an important factor for regulating the development of CS. The consequences of its dysfunction make an important contribution to the development of multiple organ failure.

Among patients with sepsis, the presence of CD64+ neutrophils in the circulation was associated with a poor prognosis [78]. The CD64 molecule is the FC gamma receptor 1 (FcγR1), which is typically expressed on monocytes [79]. It is currently unclear whether CD64+ neutrophils can be observed in CS or at least in some of its phenotypes.

## 8. Lymphocytes in Myocardial Infarction-Associated Shock

Myocardial infarction-induced damage is accompanied by the activation of the adaptive immune response, which is primarily regulated by T lymphocytes (Figure 3). In the absence of CD4+ T lymphocytes, myocardial recovery is impaired and HF advances [80].

The neutrophil-to-lymphocyte ratio (NLR) assessed on admission appeared to be important. An elevated NLR was identified as a significant predictor of 30-day mortality in patients with MI CS [81]. Furthermore, patients with more severe SCAI shock stage demonstrated a corresponding increase in the NLR [10]. Another surrogate biomarker for assessing the severity of systemic inflammation is the lymphocyte-to-monocyte ratio (LMR). Its lower values, on the contrary, were associated with the increased risk of in-hospital mortality in patients with CS [33].

The most critical factor for patients with CS is the decreased level of FoxP3+Treg responsible for regulating the strength of the immune response. Tregs express the transcription factor Forkhead box P3 (FOXP3), implement peripheral immunosuppression and maintain autotolerance [80]. In contrast to both CS survivors and MI patients, patients who died within 28 days after the onset of CS showed significantly decreased levels of Treg cells and increased levels of the subpopulation of lymphocytes with pronounced inflammatory activity, namely T helper 17 (Th17) lymphocytes [82]. A Th17/Treg ratio exceeding 0.33 predicted in-hospital mortality within 28 days with 90% sensitivity and 80% specificity [82]. On the other hand, Tregs might exert a detrimental effect on microcirculation and cerebral perfusion: they potentiate the interaction between the adhesion molecules ICAM-1 on vascular endothelium and LFA-1 on platelets, promoting thrombosis development [83].

## 9. Clonal Hematopoiesis in Cardiogenic Shock

The phenomenon of clonal hematopoiesis (CH) of indeterminate potential is currently regarded as an important factor in the pathogenesis of a number of diseases, including MI, HF, and CS [84]. CH involves the acquisition of somatic mutations of potentially oncogenic genes in hematopoietic stem cells. The risk of malignancy associated with this process is minimal (estimated at 1% per year). However, clones of immune cells are formed, exhibiting functional abnormalities [6]. The incidence of CH increases with age and is typically associated with the genes DNMT3A (DNA methyltransferase 3A), TET2 (TET methylcytosine dioxygenase type 2), ASXL1 (ASXL transcription regulator 1), and JAK2 (Janus kinase 2) [85]. CH carriers among patients with CS were characterized by a more pronounced decline in renal function, elevated levels of NT-proBNP, IL-6 and lactate in the circulation, and a worse prognosis compared to CS patients without CH [84]. A retrospective study demonstrated that CH in the TET2 gene was associated with an increased concentration of the secretory form of the CD40 (sCD40), IFN-γ, IL-4, TNF and an increased mortality rate in CS patients on days 30, 90 and after 3 years of follow-up [86].

## 10. Immunomodulatory Therapy in Cardiogenic Shock

The current therapeutic approaches in patients with MI CS include a non-specific blockade of inflammation in vivo, sorption technologies or targeted influence on inflammatory mediators (cytokines or bioactive substances) through the blockade with monoclonal antibodies (Figure 4).

Despite the obvious role of inflammation in CS, the results of immunomodulatory therapy in patients with CS remain controversial (Table 2).

The TRIUMPH study was focused on the non-selective nitric oxide synthase inhibitor L-N-monomethylarginine (L-NMMA). However, the assessment of 30-day mortality revealed no reduction in this indicator [87]. In the ACCOST-HH study, patients received monoclonal antibodies to adrenomedulin, which, being released in the interstitial space of blood vessels, contributes to inflammation development. The study also revealed no reduction in 30-day mortality in patients receiving adrenomedulin antibodies compared to placebo [88].

The COCCA (low-dose corticosteroids for CS in adult patients) study has been initiated and is currently underway. In this study, patients receive low doses of glucocorticoids (hydrocortisone and fludrocortisone) to control SIRS. The aim is to evaluate patient survival on days 7, 28 and 90 of follow-up [53].

The IMICA Trial study (treatment effects of IL-6 receptor antibodies for modulating the systemic inflammatory response after out-of-hospital cardiac arrest) yielded significant results regarding the efficacy of monoclonal antibodies targeting IL-6 (tocilizumab) in mitigating the systemic inflammatory response as well as the extent of myocardial and cerebral damage in patients with cardiac arrest [89]. A single dose of tocilizumab was observed to significantly reduce systemic inflammation, reducing myocardial injury [89]. Nevertheless, the efficacy of this approach in patients with CS remains uncertain as does the impact of IL-6 blockade on short- and long-term outcomes in this cohort of patients.

An attempt to eliminate the cumulative risk associated with the cytokine storm in patients with CS is realized through the use of sorption approaches [11,91]. Cytokine sorption is carried out along with ECMO and shows promising results: a decreased need for inotropes, the restoration of normal blood cell counts, and the successful mitigation of CS, despite the presence of secondary bacterial infection [91]. Although the data on CS are still at the clinical study level, this direction is promising, albeit with some contradictions. Retrospective studies of patients with sepsis and COVID-19 demonstrated the efficacy of the combined use of ECMO and hemoadsorption in reducing the severity of systemic inflammation [92]. At the same time, the results of the only randomized prospective study were disappointing: in patients with COVID-19, a sorption column inserted into the ECMO circuit decreased IL-6 levels significantly, but it was associated with increased 30-day mortality [90]. The study included only 34 patients randomized into two subgroups and involved only patients with COVID-19; therefore, there is no compelling reason to reject the benefits of using sorption technologies in patients with CS.

Another approach to anti-inflammatory therapy is the use of CRP apheresis in CS. To date, only one small-scale study has been conducted using this therapeutic intervention. CRP levels were observed to decrease significantly after apheresis, and the outcomes in patient survival were found to be encouraging. The data indicated a 100% survival rate after 30 days of follow-up with an adverse outcome documented in a single patient after 8 months [63]. Further studies are required to validate the efficacy of CRP apheresis in patients with CS in randomized controlled trials.

## 11. Directions for Future Research

Given the important role of inflammation in the pathogenesis of CS and recent advancements in anti-inflammatory therapy in patients with CS, it is critical to identify the group of patients who could benefit most from inflammation control.

The question arises about the criteria to be used for inflammation phenotyping in patients with CS. It is preferable to use basic clinical parameters that can be readily determined in routine practice when making critical decisions. For example, the prognosis for patients with COVID-19 was assessed by the CRP level, which demonstrated high prognostic significance (OR: 5.78, 95% CI 2.86-11.63) [93]. However, clinical practice employs more complex scales based on the analysis of inflammatory markers (IL-6, IL-8, soluble TNF receptor (sTNFR1), intercellular adhesion molecule-1 (ICAM-1)), biochemical parameters (bilirubin, bicarbonate, albumin, glucose) and parameters of the hemostasis/fibrinolysis system (plasminogen activator inhibitor-1 (PAI-1), von Willebrand factor, and protein C) [94].

Systemic inflammation indices obtained based on the complete blood count data become increasingly prevalent in assessing prognosis [95]. Among patients with acute coronary syndrome who underwent percutaneous coronary intervention, the indices SIRI (systemic inflammation response index), AISI (aggregate index of systemic inflammation) and NLR (neutrophil-to-lymphocyte ratio) are effective in assessing prognosis regardless of their gender, age, smoking status, dyslipidemia, hypertension, diabetes mellitus, and heart failure [96]. NLR showed considerable sensitivity in assessing prognosis in patients with CS [81]. However, an alternative study indicated that SOFA (sequential organ failure assessment) and SAPSII (simplified acute physiology score II) indices, based on clinical data, exhibit superior sensitivity and specificity [97]. It should be noted that all previous studies were observational and did not evaluate the effect of inflammatory phenotypes on the effectiveness of anti-inflammatory therapy, which is the subject of further scientific and practical investigation.

In addition, the determination of key biological factors that underpin each inflammatory phenotype in MI CS, with the identification of endotypes, would facilitate the development of targeted therapeutic strategies for each patient [94]. An endotype may be described as “a subtype of a disease defined by a distinctive leading pathophysiological mechanism” [98]. Currently, the identification of endotypes is widely used in internal and infectious diseases. For example, several approaches have been proposed for the classification of asthma, including identification of the main driving force of inflammation. Neutrophilic and eosinophilic asthma may be revealed, which should be considered during the choice of therapeutic approach [99]. The exploration of inflammatory endotypes in MI prompted the idea that they may resemble those observed in sepsis and represent both the activation of inflammatory response (patients were characterized by an upregulation of IL-1β gene expression) and immune suppression (patients were characterized by decreased endotoxin response) [57], which may explain unsuccessful attempts to improve the prognosis of MI patients targeting inflammation. The identification of MI CS endotypes requires comprehensive cellular and molecular profiling, genetic and epigenetic studies, artificial intelligence technologies, and linking of the revealed endotypes to the disease outcomes [100,101]. Nevertheless, it seems plausible that this may emerge in the near future due to the accelerating pace of technological advancement and will lead to the development of personalized treatment approaches in CS patients.

The identification of the optimal timing and volume of anti-inflammatory therapy is a significant challenge in managing patients with MI CS. Since inflammation is a prerequisite for the full restoration of myocardial function [20], its absolute suppression in the early stages is undesirable and potentially dangerous. It is conceivable that precision medicine methodologies will provide a resolution to this question in the future.

## 12. Conclusions

In conclusion, this review has demonstrated that inflammation and immune system dysregulation play a pivotal role in the pathogenesis of MI CS and associated multiple organ dysfunctions. Both cells of the innate and adaptive immune system are being involved; and very often, aseptic inflammation during MI is accompanied by the infections as MI CS progresses. The current anti-inflammatory strategies to target inflammation during MI CS are represented by specific as well as non-specific approaches, and they include low-dose glucocorticoids, monoclonal antibodies, specific blockers of various inflammatory pathways and hemoadsorption. To further develop the spectrum and efficiency of anti-inflammatory medications for an extremely severe and heterogeneous group of patients with MI CS, it is first necessary to identify the patient group that would benefit the most from these therapeutic measures, identify the most promising diagnostic and therapeutic targets, and determine the time interval for justified anti-inflammatory therapy.

## Figures and Tables

**Figure 1 biomedicines-12-02073-f001:**
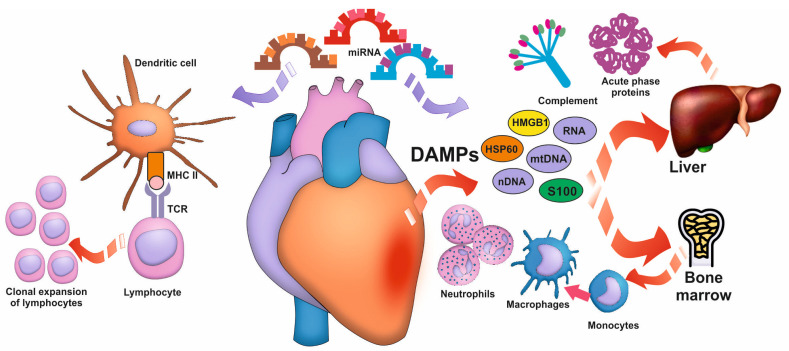
Development of inflammation in myocardial infarction. Myocardial injury results in the release of DAMPs from the injured myocardium that cause inflammation through the activation of the complement system, the acute-phase response, and the recruitment of innate immune cells (neutrophils, monocytes, macrophages, and dendritic cells). This is followed by the recruitment of adaptive immune cells (lymphocytes). Moreover, microRNAs are involved in regulating the inflammatory response and its potency. Data from Frangogiannis (2014) [27]; Hofmann et al. (2015) [28]; Li et al. (2016) [29]; Timmermans et al. (2016) [22]; Pluijmert et al. (2021) [25]; Anzai et al. (2022) [23]; Varzideh et al. (2022) [30]; and Zhang et al. (2022) [31]. The icon of bone marrow is created by kerismaker and was downloaded from www.flaticon.com (https://www.flaticon.com/free-icon/bone-marrow_3816804?term=bone+marrow&page=1&position=10&origin=search&related_id=3816804, accessed on 18 August 2024). DAMPs, damage-associated molecular patterns; HMGB1, high-mobility group protein B1; HSP, heat shock protein; MHC II, major histocompatability complex II; miRNA, micro-ribonucleic acid; mtDNA, mitochondrial deoxyribonucleic acid; nDNA, nuclear deoxyribonucleic acid; RNA, ribonucleic acid; S100, S100 protein; TCR, T-cell receptor.

**Figure 2 biomedicines-12-02073-f002:**
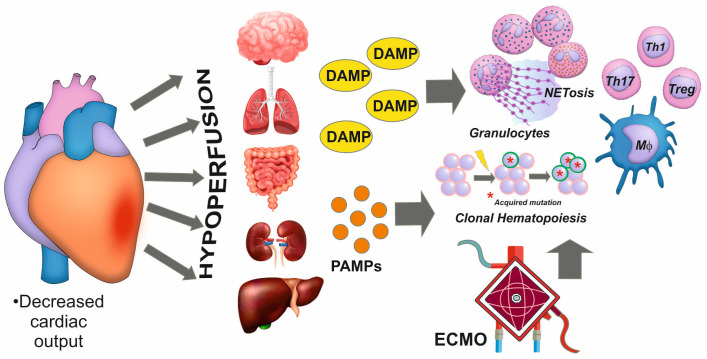
Inflammation development in cardiogenic shock. The decreased cardiac output is associated with hypoperfusion and organ damage. In this case, an increase in damage-associated molecular patterns (DAMPs) is accompanied by an increase in pathogen-associated molecular patterns (PAMPs), which cause the activation and subsequent dysfunction of immune cells. NETosis, the release of granulocyte nucleic acids into the extracellular environment, and clonal hematopoiesis associated with somatic mutations of hematopoietic stem cells are observed. Resuscitation procedures such as extracorporeal membrane oxygenation (ECMO) can also stimulate the dysfunction of immune cells. DAMPs, damage-associated molecular patterns; ECMO, extracorporeal membrane oxygenation; Mφ, macrophage; PAMPs, pathogen-associated molecular patterns; Th, T helper lymphocyte.

**Figure 3 biomedicines-12-02073-f003:**
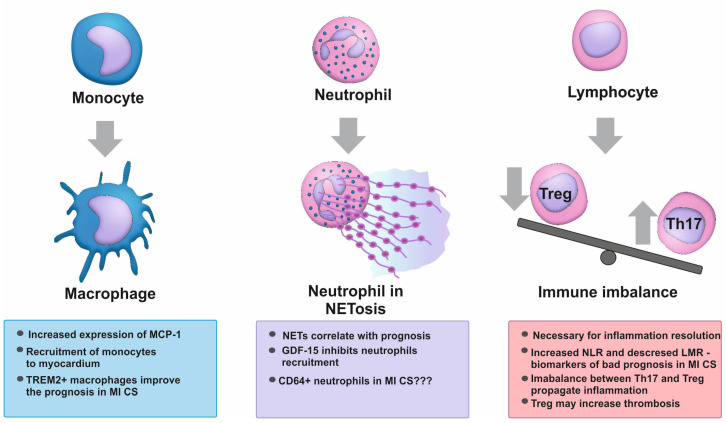
The role of immune cells in the development of inflammation during myocardial infarction-associated shock. CD, cluster of differentiation; GDF-15, growth-differentiation factor 15; LMR, lymphocytes-to-monocytes ratio; MCP-1, monocyte chemotactic protein-1; MI CS, myocardial infarction-associated shock; NET, neutrophil extracellular trap; NLR, neutrophils-to-lymphocytes ratio; Th, T helper lymphocyte; Treg, regulatory T lymphocytes; TREM2, triggering receptor expressed on myeloid cells 2.

**Figure 4 biomedicines-12-02073-f004:**
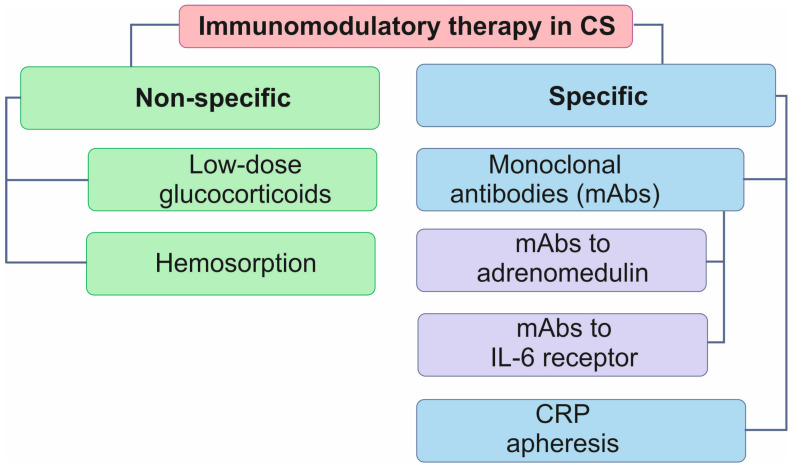
Modern approaches to control inflammation in cardiogenic shock. CRP, C-reactive protein; CS, cardiogenic shock; IL, interleukin; mAbs, monoclonal antibodies.

**Table 1 biomedicines-12-02073-t001:** Clinical studies investigating inflammation in cardiogenic shock.

Authors	Patients with CS	Study Points	Results
Cuinet et al. (2020) [51]	24 patients (12 cases of ACS; 6 cases of chronic cardiomyopathy)	Days 1, 3, 6–8 after CS development	Day 1: ↑ neutrophils, IL-6, IL-10, MCP-1, eotaxin; Day 3: ↓ lymphocytes, monocytes; Days 6–8: ↓ lymphocytes, monocytes, ↑ eosinophils. Infectious complications 62%
Kunkel et al. (2023) [52]	1716 patients with acute MI who survived within 48 ± 12 h on admission to hospital	Retrospective study	Increased 30-day mortality in patients with high levels of hsCRP and leukocytes
Mekontso Dessap et al. (2024) [53]	406 patients with CS (134 cases of CS of ischemic etiology; 65 cases of CS after sepsis)	Retrospective study	Increased mortality after 1 month and after 1 year of follow-up in patients with elevated CRP level (greater than 69 mg/L) on admission
Jentzer et al. (2020) [3]	8999 cardiac intensive care patients	Retrospective study	Increased short- and long-term mortality in patients with the SIRS (≥2/4 of the following criteria: elevated heart rate >90/min, elevated body temperature >38 °C, elevated total white blood cell count >12 × 10^9^/L, elevated respiratory rate >20/min)
Dettling et al. (2024) [54]	1116 patients with CS according to ICD-10 (including 530 patients with acute MI)	Retrospective study	Increased 30-day in-hospital mortality with elevated CRP level above the median value (17 mg/L) in the absence of MCS measures
Diakos et al. (2021) [55]	134 patients with CS (73 patients with acute MI) and MCS (ECMO or Impella)	Retrospective study	Lower baseline neutrophil-to-lymphocyte ratio in survived patients; lower IL-6 level after MCS in survived patients
Parenica et al. (2017) [5]	80 patients with acute MI	Time of admission, 12 h, 24 h, 48 h, 72 h, 96 h, 7 days after admission, 3 months	Elevated levels of CRP, procalcitonin, presepsin, and pentraxin-3 in the first 24 h of follow-up are predictors of 3-month mortality in patients with CS
Peters et al. (2024) [56]	111 patients with STEMI after early PCI	Days 1, 2, 3, 30, and 60 after CS development	Elevated adrenomodulin on the first day of CS indicates the degree of multiple organ dysfunction and is a predictor of mortality in patients with CS, in contrast to patients with sepsis

ACS, acute coronary syndrome; CRP, C-reactive protein; CS, cardiogenic shock; ECMO, extracorporeal membrane oxygenation; ICD, International Classification of Diseases; IL, interleukin; MI, myocardial infarction; MCS, mechanical circulatory support; PCI, percutaneous coronary intervention; SIRS, systemic inflammatory response syndrome; STEMI, ST segment-elevation myocardial infarction; ↑ indicates increase of the parameter; ↓ indicates decrease of the parameter.

**Table 2 biomedicines-12-02073-t002:** Clinical trials targeting inflammation in intensive care.

Trial	Therapeutical Approach	Sample Size	Effect
TRIUMPH [87]	L-NMMA, non-selective inhibitor of nitric oxide synthase	658 patients with MI CS	No effect
ACCOST-HH [88]	Monoclonal antibodies to adrenomedulin	150 patients with CS	No effect
IMICA [89]	Antibodies to IL-6 receptor	80 patients with out-of-hospital cardiac arrest	Reduction in inflammation and myocardial injury
CYCOV [90]	Combination of ECMO and cytokine adsorption therapy	34 patients with severe COVID-19 pneumonia	Negative effect on survival; no reduction in inflammation
COCCA [53]	Hydrocortisone and fludrocortisone in low doses	380 patients with CS (planned)	Ongoing

COVID, coronavirus disease; CS, cardiogenic shock; ECMO, extracorporeal membrane oxygenation; IL, interleukin; L-NMMA, L-N-monomethylarginine;.

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
