# Peer review of "The Role of Inflammation in the Pathogenesis of Cardiogenic Shock Secondary to Acute Myocardial Infarction: A Narrative Review"

_biomedicines, 2024, doi:10.3390/biomedicines12092073_

Round 1

Reviewer 1 Report

Comments and Suggestions for Authors

1) Sections 5-7 require an illustration 

2) Figures 1, 2 require explanations. Abrevations... 

3) Figure 2 has a designation in Russian

4) Directions for future research. Should be shortened

Comments on the Quality of English Language

Additional grammar and punctuation checking is required.

Author Response

Dear reviewer, we thank you for your time and efforts that you spent working with our manuscript. We are highly grateful for your valuable comments and tried our best to address them in the revised version of the paper. The revisions are described below, and are marked with yellow colour through the manuscript:

1) Sections 5-7 require an illustration 

We have added Figure 3. The role of immune cells in the development of inflammation during myocardial infarction-associated shock

2) Figures 1, 2 require explanations. Abrevations... 

We have deciphered abbreviations, presented in all the figures

3) Figure 2 has a designation in Russian

We have corrected the letter into the Greek Ï•

4) Directions for future research. Should be shortened

We have excluded the deciphering of the inflammatory indices. We left only those, which appeared to be valuable to predict the prognosis.

The section, devoted to endotypes, was extended though, as it was requested by another reviewer.

5) Comments on the Quality of English Language

Additional grammar and punctuation checking is required

We have consulted a professional translating service

Reviewer 2 Report

Comments and Suggestions for Authors

This review addresses the critical issue of cardiogenic shock (CS) as a severe complication of myocardial infarction (MI), emphasizing the high mortality rates associated with it. The paper explores the role of systemic inflammatory response in the pathogenesis of MI CS, describes the primary processes from MI onset to multiple organ dysfunction, and discusses current and potential anti-inflammatory therapies. The topic seems highly relevant given the high mortality rates associated with MI CS. Understanding the inflammatory mechanisms can lead to better therapeutic strategies in clinics.

Some important comments:
1. Lines 65-67: This idea should be supported by some references: "there is no understanding of how septic and aseptic components contribute to the development and progression of shock; markers have not been identified to distinguish aseptic inflammation from septic inflammation in individuals with comorbid pathology."

2. Recent research increasingly highlights inflammaging as a critical factor in the progression of various diseases [pubmed.ncbi.nlm.nih.gov/37047346/ ] [pubmed.ncbi.nlm.nih.gov/36611900/ ]. I recommend specifically addressing the role of inflammaging in the context of myocardial infarction.

3. Figure 1 is very primitive. I recommend redrawing it. I also recommend including additional visual aids such as flow charts, diagrams, and tables to summarize key points, processes, and therapeutic approaches.

4. Lines 286-291: References are required.

5. Lines 106, 143, 150, 251, 342: Since the authors discussed and distinguished between apoptosis and necrosis, it would be beneficial to address other forms of cell death as well.

6. Line 209: Check the correctness of this sentence: "This is due to their capacity to potentiate the interaction between adhesion molecules ICAM-1 on vascular endothelium and LFA-1 on platelets, thereby promoting thrombosis development".

7. I recommend including more data on how specific inflammatory phenotypes and endotypes are identified, along with examples. This can help in understanding the heterogeneity of MI CS types.

8. Conclusion section is poorly written. Lines 520-524 are entirely superfluous. I recommend including only *insights* that the authors find particularly valuable based on their review.

Author Response

Dear reviewer, we thank you for your time and efforts that you spent working with our manuscript. We are highly grateful for your valuable comments and tried our best to address them in the revised version of the paper. The revisions are described below, and are marked with yellow colour through the manuscript:

  1. Lines 65-67: This idea should be supported by some references: "there is no understanding of how septic and aseptic components contribute to the development and progression of shock; markers have not been identified to distinguish aseptic inflammation from septic inflammation in individuals with comorbid pathology."

We have added the links and mentioned that we are speaking of the specific markers.

  1. Recent research increasingly highlights inflammaging as a critical factor in the progression of various diseases [pubmed.ncbi.nlm.nih.gov/37047346/ ] [pubmed.ncbi.nlm.nih.gov/36611900/ ]. I recommend specifically addressing the role of inflammaging in the context of myocardial infarction.

We have addressed the role of inflammaging in myocardial infarction in lines 190-202

  1. Figure 1 is very primitive. I recommend redrawing it. I also recommend including additional visual aids such as flow charts, diagrams, and tables to summarize key points, processes, and therapeutic approaches.

We have changed Figure 1. We have also created Figure 3 (The role of immune cells in the development of inflammation during myocardial infarction-associated shock), Figure 4 (Modern approaches to control inflammation in cardiogenic shock) and Table 2 (Clinical trials targeting inflammation in intensive care).

  1. Lines 286-291: References are required.

We have added references here (Lines 335-337).

  1. Lines 106, 143, 150, 251, 342: Since the authors discussed and distinguished between apoptosis and necrosis, it would be beneficial to address other forms of cell death as well.

We have added a fragment describing various type of cell death in myocardial infarction (Lines 107 - 114).

  1. Line 209: Check the correctness of this sentence: "This is due to their capacity to potentiate the interaction between adhesion molecules ICAM-1 on vascular endothelium and LFA-1 on platelets, thereby promoting thrombosis development".

We have changed the sentence: “On the other hand, Tregs might exert a detrimental effect on microcirculation and cerebral perfusion: they potentiate the interaction between the adhesion molecules ICAM-1 on vascular endothelium and LFA-1 on platelets, thereby promoting thrombosis development”

  1. I recommend including more data on how specific inflammatory phenotypes and endotypes are identified, along with examples. This can help in understanding the heterogeneity of MI CS types.

We have expanded the fragment devoted to endotypes in Lines 567 – 578.

  1. Conclusion section is poorly written. Lines 520-524 are entirely superfluous. I recommend including only *insights* that the authors find particularly valuable based on their review.

We have edited this section and excluded the above mentioned lines.

Reviewer 3 Report

Comments and Suggestions for Authors

Thank you for inviting me to review the article entitled ‘The role of inflammation in the pathogenesis of cardiogenic shock secondary to acute myocardial infarction.

Cardiogenic shock (CS) represents the most severe manner of  acute heart failure (AHF), defined as a low cardiac output condition due to cardiac dysfunction, leading to severe end-organ hypoperfusion associated with tissue hypoxia and increased lactate levels. In-hospital mortality of patients with CS is in the range of 30-50%. The current article addresses an important aspect of CS pathogenesis.

Title: It is worth adding that this is an review paper.

Abstract: It represents the work done well.

Introduction: It introduces the reader to the subject well.

Methods: It is worth stating what type of review it is. The selection of articles should take place in accordance with current recommendations. For a systematic review, the principles of PRISMA 2020 guidelines should be applied.

General: The authors described in detail the mechanisms of the inflammatory substrate of CS.

The authors demonstrated their knowledge of the subject.

The article is worth publishing once it has been corrected.

Author Response

Dear reviewer, we thank you for your time and efforts that you spent working with our manuscript. We are highly grateful for your valuable comments and tried our best to address them in the revised version of the paper. The revisions are described below, and are marked with yellow colour through the manuscript:

  1. Title: It is worth adding that this is an review paper.

We have corrected the title: The role of inflammation in the pathogenesis of cardiogenic shock secondary to acute myocardial infarction: a narrative re-view

  1. Methods: It is worth stating what type of review it is. The selection of articles should take place in accordance with current recommendations. For a systematic review, the principles of PRISMA 2020 guidelines should be applied.

We have mentioned that it was a narrative review. But added “Methods” chapter anyway, describing our research approach (Lines 91-104).

Round 2

Reviewer 1 Report

Comments and Suggestions for Authors

The authors have taken my comments into account. The article can be accepted in its current form.

Reviewer 2 Report

Comments and Suggestions for Authors

The authors satisfactorily addressed all my comments and made the necessary revisions to the manuscript.